# Discrimination of Red Wines with a Gas-Sensor Array Based on a Surface-Acoustic-Wave Technique

**DOI:** 10.3390/mi10110725

**Published:** 2019-10-26

**Authors:** Min-Han Lin, Ling-Yi Ke, Da-Jeng Yao

**Affiliations:** 1Institute of Nanoengineering and Microsystems, National Tsing Hua University, Hsinchu 30013, Taiwan; c04aup6xup6@gmail.com; 2Department of Power Mechanical Engineering, National Tsing Hua University, Hsinchu 30013, Taiwan; lingyi0412@gmail.com

**Keywords:** surface acoustic wave, aroma of red wines, polymers

## Abstract

We applied a thermal-desorption gas-chromatograph mass-spectrometer (TD-GC–MS) system to identify the marker volatile organic compounds (VOCs) in the aroma of red wine. After obtaining the marker VOC, we utilized surface acoustic waves (SAWs) to develop a highly sensitive sensing system as ‘electronic nose’ to detect these marker VOC. The SAW chips were fabricated on a LiNbO_3_ substrate with a lithographic process. We coated sensing polymers on the sensing area to adsorb the marker VOC in a sample gas. The adsorption of the marker VOC altered the velocity of the SAW according to a mass-loading effect, causing a frequency decrease. This experiment was conducted with wines of three grape varieties—cabernet sauvignon, merlot and black queen. According to the results of TD-GC–MS, the King brand of red wine is likely to have unique VOC, which are 2-pentanone, dimethyl disulfide, 2-methylpropyl acetate and 2-pentanol; Blue Nun-1 probably has a special VOC such as 2,3-butanedione. We hence used a SAW sensor array to detect the aroma of red wines and to distinguish their components by their frequency shift. The results show that the use of polyvinyl butyral (PVB) as a detecting material can distinguish Blue Nun-2 from the others and the use of polyvinyl alcohol (PVA) can distinguish King from the others. We conducted random tests to prove the accuracy and the reliability of our SAW sensors.

## 1. Introduction

Red wine has its own unique aroma; according to some authorities 85% of the taste of red wine is sensed by olfaction. Much effort has been devoted to analyze the aroma of red wines to know their composition; this aroma is composed of many volatile organic compounds (VOCs). From other work, we know that these VOCs determine the characteristics of red wine aroma; some VOCs, such as vanillin or β damascenone or (*E*)-β-methyl-γ-octalactone, have a positive correlation with a pleasant aroma, whereas VOC such as 4-ethyl- and vinyl-phenols or 3-(methylthio)-1-propanol or phenylacetaldehyde have a negative correlation, being associated with an unfavorable odor [1]. We can utilize these unique VOC in the aroma of red wine for differentiation and to ensure the accuracy of information such as the origin and the grape variety to protect consumers. Based on much research, the aroma of a red wine is affected by many complicated factors including the grape variety [2] and the production process (crushing, maceration [3], fermentation [4], pressing and aging [5]). Among these factors, the grape variety is the essential key to the flavor of red wine. In this research, we sought to explain the variation of aroma composition among the grape varieties. We hence chose the most common grape varieties (cabernet sauvignon and merlot) and a unique variety (black queen) of Taiwan to undertake an analysis of their aromas.

As a traditional and still current analytical method, a wine taster with adequate training can identify red wines and derive some information such as year and grape variety through the human olfactory sense and taste, but analysis of this kind is susceptible to inaccuracy. At present, in seeking detailed and accurate information, inspection personnel must use a specific machine to analyze the composition of the aroma of a red wine, but this method requires much time. An analysis of aroma characteristics thus involves recording signals generated from gaseous mixtures (like with a human nose) and then comparing the reaction patterns produced by the various samples. These methods have been widely used in wine analysis, including quality control [6], but, because of the complexity of the sample, the small differences in composition between various wines and the presence of water and ethanol, which interfere with the sensor response, wine analysis is a problem that lacks full resolution [7]. For this reason, efforts are exerted to develop sensor arrays with improved sensitivity and reproducibility. Developing a new method that is convenient, cheap and rapid hence becomes an important task. A portable gas sensor is an appropriate instrument for the detection and analysis of red wine. As Table 1 [8] shows, many commercial gas sensors have been developed for a specific gas based on varied working mechanisms. Among them, a metal-oxide-semiconductor gas sensor is widely used because a sensor of this kind possesses advantages of great stability and modest cost. In much research, a metal-oxide-semiconductor gas-sensor array served to detect and to discriminate red wines [9]. This sensor has, however, a serious problem in that it must operate at a high temperature. Here we compare the advantages and disadvantages of some commercial sensors. An infrared gas sensor [10] is effective for precise detection but it is easily perturbed by temperature and humidity; the device components are expensive. A quartz-crystal microbalance has great precision and sensitivity, but its range of detection is inadequate for detection of a gas at a small concentration [8]. A metal-oxide-semiconductor field-effect transistor (MOSFET) gas sensor has advantages of small dimension and a modest cost of production but many shortcomings, such as variation of sensitivity with temperature and humidity, a short lifetime and time-consuming operation [11]. These gas sensors are hence inconvenient and unsuitable for the detection of red wines because of the specified shortcomings. An electronic-nose system composed of a surface-acoustic-wave sensor array that can imitate a human nose to identify the aroma of red wines is an appropriate selection for testing to detect a red wine because its advantages are a modest cost, effective sensitivity to various chemicals, small power consumption, small size, satisfactory repeatability and great sensitivity [12,13,14,15].

The main purpose of this research comprised two parts. The first part was to apply a system of a thermal-desorption gas chromatograph and a mass spectrometer (TD-GC–MS) to analyze the aroma of red wine made from three grape varieties (cabernet sauvignon, merlot and black queen) to determine the unique volatile organic compounds (VOC) in specific aromas that serve as markers, so to implement differentiation. The second part was to develop an electronic-nose (e-nose) red-wine differentiation system; for this purpose, we utilized a SAW sensor coated with varied polymer films to form a sensor array. The chemical properties of a polymer-sensing film should be as diverse as possible, so as to provide the e-nose system with an ability to undertake an analysis of analytes over the largest possible range [16].

Aromatic compounds typically have small molecular masses ranging between 30 and 300 Da (g/mol) [17]. In our research, we hence chose a SAW gas sensor to measure the aroma of a red wine because of its appropriate limit of detection (LoD). Our application of a SAW gas sensor array to quantify wines of diverse grape varieties demonstrated satisfactory results.

## 2. Materials and Methods

### 2.1. Red Wine Samples

In this research, four red wines were chosen for analysis: King from LH winery in Taiwan, Blue Nun-1 (2014), Blue Nun-2 (2014) and Dadung (2004), as shown in Table 2. The Dadung wine sample is described as a mixture of black queen and cabernet sauvignon. King was made from black queen, Blue Nun-1 from cabernet sauvignon and Blue Nun-2 from merlot. 

### 2.2. Sample Extraction

Before admitting the aroma of red wine into the TD-GC–MS, we extracted the sample gas into a tube that contained an adsorption sorbent (TENAX-TA, PerkinElmer, Waltham, MA, USA) to adsorb the VOC. The tube was then sent into the thermal-desorption (TD) system to execute a two-stage thermal-desorption step to desorb the aroma of red wine and to enhance the concentration of the aroma before being sent into the gas chromatograph–mass spectrometer (GC–MS) system. To extract the aroma of red wine, we poured red wine (1 mL) into a glass container, then used a micropump (MicroJet Technology, Hsinchu, Taiwan) to extract red wine aroma for 30 min. The adsorption sorbent inside the tube trapped the VOC in the red wine aroma, as shown in Figure 1. After 30 min, we placed the tube into the TD, and simultaneously placed the glass vessel that contained the red wine into the SAW sensing system to detect its aroma.

### 2.3. Thermal Desorption-Gas Chromatograph–Mass Spectrometer (TD-GC–MS) 

In this research, a gas chromatograph (Agilent 7890A, Agilent, Santa Clara, CA, USA) and a mass spectrometer (Agilent 5975C, Agilent) were used to analyze the aroma of red wines. After collection of the aroma, the tube was subjected to a thermal desorption in two stages; the VOC were eventually transferred into the GC–MS with a carrier gas. We selected two columns: one (J&W DB-624, Agilent) was slightly polar and had a strong affinity for VOC of small polarity; the other (J&W DB-WAX, Agilent) was highly polar. Both columns were fitted with a fused-silica capillary column (30 m, 0.25 mm and film thickness 0.25 μm). The oven temperature, initially 35 °C, was held there for 6 min. The temperature was then increased to 135 °C at 3 °C/min and eventually increased to 230 °C at 6 °C/min, held there for 5 min. In this process, we used pure helium (99.9995%) as a carrier gas to flow through the tube at rate 1 mL/min. The temperature of both the mass spectrometer (MS) transfer line and the ion source was 230 °C.

### 2.4. Principle of a SAW Sensor

According to the fundamental working mechanism a SAW sensor utilizes the velocity change of a surface acoustic wave propagating through the surface of a piezoelectric material. A surface acoustic wave is stimulated with a time-varying electric field that deforms the piezoelectric material through an inverse-piezoelectric effect. When gas molecules pass through the sensing area, which is a coated polymer film such as PVB or poly-4-vinylpyridine (P4VP), gas molecules form a weak van der Waals bond with the polymer film, thus becoming adsorbed on that film and causing a velocity change [18].

The velocity change, which is also a frequency shift, arises from three effects—a mass-loading effect, elastic properties and an acoustoelectric effect; the correlation equation follows:(1)Δff0≅Δϑϑ0=−cmf0Δ(mA)+4cef0Δ(hG′)−k22(11+ϑ02Δ(csσs)2).

Here, the frequency shift is majorly affected by the mass-loading effect because the elastic properties have little influence on the SAW sensor [19]; the acoustoelectric effect is found in mainly semiconductor materials. As Equation (1) shows, the mass is proportional to the decrease of velocity, which correlates with a decrease of frequency. When gas molecules adsorb on a sensing area, the mass hence increases, resulting in a decreased velocity and frequency [20].
(2)$$\frac{\Delta f}{f_{0}}≅\frac{\Delta \vartheta }{\vartheta_{0}}=-c_{m}f_{0}\Delta \left(\frac{m}{A}\right).$$

In Equation (2), all parameters have SI units. The last term, which is the mass-loading effect, contains $c_{m}$, $f_{0}$, *m* and *A* that represent the sensitivity coefficient of mass (m^2^/kg), the initial SAW frequency (Hz), mass (kg) and surface-sensing area (m^2^), respectively [21].

### 2.5. Design and Fabrication of SAW Sensors

In our SAW chip, we designed a positive-feedback circuit to generate a stable oscillation voltage to stimulate the SAW frequency. The SAW sensor was thus composed of an oscillator, which can generate a time-varying electric field to the piezoelectric substrate and stimulate a surface acoustic wave (SAW), and a SAW chip that was coated with various polymers. 

If we seek to stimulate a specific frequency, the design parameters including interdigital- transducer (IDT) pairs, finger width, material and thickness must be carefully considered. The IDT was patterned with lithography; gold was deposited with an e-gun evaporator. The IDT, consisting of 50 pairs of finger width 8.5 μm, was deposited on a LiNbO_3_ substrate of thickness 100 nm. The process included three main steps: (a) photolithography, (b) e-gun evaporation and (c) lift-off, as shown in Figure 2.

The LiNbO_3_ wafer was cleaned with acetone and isopropanol (IPA) to remove organic compounds, then rinsed with deionized water and eventually immersed in piranha solution (sulfuric acid:hydrogen peroxide = 7:1) for 10 min at 90 °C to remove light-metal pollution. To increase the adhesion between the wafer and the photoresist, the wafer was coated with hexamethyldisilazane (HMDS) vapor for 5 min. We then used a spin coater to coat a positive photoresist (AZ5214, Clariant, Muttenz, Switzerland) on it (thickness 1.5 µm) at 3000 rpm for 30 s. Afterwards, the wafer was baked at precisely 100 °C for 1 min to enable the solvent inside the photoresist to evaporate, before cooling to ~23 °C. The wafer was then exposed to ultraviolet light to define a pattern; the pattern was eventually developed with a photoresist developer (AZ400K:DI water = 1:6) for 25–30 s, and inspected with an optical microscope to verify the completeness of the entire structure.

(a) E-beam evaporation:

After the photolithographic process, Cr (thickness 20 nm) was deposited first on the substrate as an adhesive layer; afterward Au (thickness 100 nm) was deposited, with e-beam evaporation.

(b) Lift-off:

After the deposition, the wafer was immersed in acetone and sonicated with an ultrasonic cleaner to remove the photoresist on the unexposed area; the defined pattern then emerged. The completed wafer was diced into chips with laser cutting.

### 2.6. Polymer Preparation and Coating

Since the composition of a red wine is complicated, we must select several polymers to form the sensor arrays for gas detection. We achieved the resolution on selecting these polymers and the gaseous components of red wine with varied capabilities of physical adsorption. In this work, we selected six polymers as sensing materials: poly-N-vinylpyrrolidone (PNVP), poly-4-vinylpyridine (P4VP), polystyrene (PS), polyvinyl butyral (PVB), polyvinyl alcohol (PVA) and polymethyl methacrylate (PMMA). PVA and PVB have strong bonding; P4VP and PNVP are highly polar molecules; PS and PMMA are less prone to hydrogen bonding. Among them, PNVP and PVA are hydrophilic polymers for which the solvent is water or alcohol, whereas PS, P4VP, PVB and PMMA are hydrophobic polymers for which the solvent is tetrahydrofuran (THF). The ratio of solute to solvent is 0.1 g to 1 mL. After preparing the polymer, we used a pipette to withdraw the polymer solution (1 μL) and dipped it onto the sensing area on the chip; afterward, we put the chip on a hotplate at 90 °C for 5 min to expel the solvent.

### 2.7. SAW Sensing System

As shown in Figure 3a, four-port SAW sensors were comprised of four SAW chips and four oscillators. Among them, there was a reference chip without coated polymer, of which the purpose was to eliminate ambient influences, such as temperature, humidity and noise. On subtracting the signals from two chips, we offset these ambient influences. Furthermore, as shown in Figure 3b, the SAW sensors were put in an isolation-sensing chamber to decrease the influence from the environment. Figure 4 shows the entire experimental setup of gas sensing. The continuous detection for red wine was conducted with a gas generator, mass-flow controller (MFC), mixing chamber, timer and solenoid valve, frequency counter, power supply and computer. For gas detection, we used a sorption technique, whereby a carrier gas flows into the glass bottle loaded with a red wine (1 mL); the carrier gas attracts the red wine aroma and volatiles to become our sample gas [22]. Next, we designed a two-stage dilution to ensure complete mixing of the sample gas with a diluent gas. The first stage involved the sample gas from red wine (1 mL, 200 sccm) and the diluent gas from a carrier gas (500 sccm) being controlled with the MFC inside the gas generator; in the second stage the gas from the first stage mixed with the diluent gas that was controlled with an external MFC. In our work, dry air served as the diluent gas. A dry gas was thought of acting as a desorption gas, controlled with the external MFC. We then set adsorption and desorption durations with a timer, so that a solenoid valve could automatically switch the sample gas and the dry gas to complete the adsorption and desorption curve. In this experiment, a power supply provided 4 V to the oscillator circuit; the oscillator circuit was connected to the SAW chip. The signal was measured with a frequency counter and recorded as frequency with Labview.

## 3. Results

### 3.1. Polymer Preparation and Coating

#### 3.1.1. Chromatography Column (DB-624) Results

The results of column DB-624, shown in Table 3, reveal that 22 VOC were found in four red wines, represented as peak area; “–” signifies that a VOC was not found in that red wine. After comparison, we obtained the result that King had perhaps unique VOC including methyl acetate, 2-pentanone, dimethyl disulfide and 2-methylpropyl acetate. Sulfur dioxide was found in Blue Nun-1 and Blue Nun-2. According to the literature, sulfur dioxide is typically added to food as a preservative to prevent processed products from spoilage [23]; sulfur dioxide could hence not be regarded as a marker that could distinguish a black queen from King and Dadung. In addition, quinoline, 6-ethoxy-1,2-dihydro-2,2,4-trimethylquinoline found in Blue Nun-2 probably arose from the thermal cracking on the column because of the high temperature. In the past, Dadung was charged with using grape juice to brew wines, for which reason the aroma might be less than that of Blue Nun-1. As we see, the peak area of Dadung is less than Blue Nun-1 in the same VOC. This result can verify that Dadung perhaps uses grape juice for the fermentation and production of red wines. In summary, only methyl acetate, 2-pentanone, dimethyl disulfide and 2-methylpropyl acetate can be regarded as markers to distinguish the King from the other red wines.

#### 3.1.2. Chromatography Column (DB-WAX) Results 

In contrast with DB-624, DB-WAX is a highly polar column that has hence a strong affinity for highly polar VOC. We thus expected that a DB-WAX column might reveal other VOC that were not found with DB-624. For the DX-WAX results, Table 4 shows that 12 VOC in total were found in the four red wines. VOC including 2-pentanol, 2,3-butanedione and 2-pentanone were found also in DB-624, verifying that they might serve as markers.

The results reveal that King had 2-pentanol as a marker. Our results indicated that Blue Nun-1 has 2,3-butanedione that might be regarded as a marker VOC. 2,3-Butanedione can hence serve as a marker VOC that can differentiate cabernet sauvignon from merlot. Moreover, Dadung had the least VOC whether on DB-624 or on DB-WAX.

In summary, on combining the results from DB-624 and DB-WAX, King had marker VOC including methyl acetate, 2-pentanone, dimethyl disulfide, 2-methylpropyl acetate and 2-pentanol. Blue Nun-1 had 2,3-butanedione that could serve as the marker VOC.

### 3.2. Polymer Preparation and Coating

#### 3.2.1. Results of a SAW Sensor Array Coated with Six Polymers

After determining the marker VOC, we used a SAW sensing system coated with PVB, PVA, PS, PNVP, P4VP and PMMA on the sensing area to detect the red wine. Among them, because the marker VOCs, which could discriminate King from Blue Nun-2 and Blue Nun-1, were highly polar compounds, we selected highly polar polymers such as PVA, PNVP and P4VP that have a strong affinity based on the van der Waals force. PVA is likely the most suitable polymer to discriminate these red wines because PVA generates hydrogen bonds to adsorb the 2-pentanone marker VOC. In the same way, we selected slightly polar polymers including PVB, PS and PMMA as sensing materials because 2,3-butanedione, which is the maker VOC of Blue Nun-2, can adsorb on these polymers depending on the van der Waals force. Based on the principle of the SAW device, the polymer adsorbs the compounds in the admitted sample gas and decreases the velocity of the surface acoustic wave, leading to a frequency decrease. Each polymer has a distinct affinity for the aroma of a red wine and causes a distinct frequency-shift finger print. In the same way, when dry gas passed through the sensing area, the compounds desorbed so that the frequency shifted back to the baseline, which means that the frequency shift equaled zero. For example, Figure 5 shows the use of PVA as a sensing material to detect four red wines. Each experiment was undertaken with three cycles of adsorption and desorption; we averaged three peak values of frequency shift to obtain Figure 6. We conducted experiments in triplicate on each red wine.

Figure 6 shows triplicated results of the average frequency shift of SAW sensors coated with six polymers. In Figure 6, we see that the PVB polymer had a large frequency shift, which could distinguish Blue Nun-2 from King, Blue Nun-1 and Dadung. Corresponding to the TD-GC–MS results, marker VOC 2,3-butanedione had perhaps a strong affinity for PVB. PVB had a slight polarity that readily captured a slightly polar VOC such as 2,3-butanedione by a van der Waals force. The PVA polymer could distinguish King from Dadung, Blue Nun-1 and Blue Nun-2 because PVA had an –OH functional group that could bind with 2-pentanol, which is a marker VOC, depending on a hydrogen bond. When we selected PVB, PS and PMMA as sensing materials, the results revealed that the frequency shift of Dadung red wine was the least because of the least VOCs that were detected, corresponding to the TD-GC–MS result. Moreover, in our triplicated experiment, we obtained that the frequency shift of the first time was the largest. As the experiment proceeds, the frequency shift of all SAW sensors coated with the six polymers gradually decreased because of the dissipation of the red wines, but this phenomenon did not influence the results from the SAW sensors.

#### 3.2.2. Blind Test of Red Wines and Lifetime of a Polymer Film

After conducting triplicated experiments, we were confident that our SAW sensors could discriminate the grape varieties including black queen, cabernet sauvignon and merlot. To increase the credibility of the SAW sensor, we undertook a blind test of red wines. Since we must consider the decay of the polymer film, we conducted the experiment on the first, fourth and seventh days. Figure 7a for the first day shows that No.1 had the greatest frequency response with PVB as the sensing material; according to our experience above, we deduced that No.1 was Blue Nun-2. When using PVA as a sensing material, No.2 had the greatest frequency shift; No. 2 was hence regarded as the King red wine. No. 3 had the smallest frequency shift when using these three polymers; No.3 is the Dadung red wine and No.4 is the Blue Nun-1. As a result, we conducted successfully a blind test of red wines. 

Four days afterward, we undertook the same experiment with the same polymer film; the result of the fourth day was that No.1 was King because of the greatest frequency shift with the PVA polymer. No. 2 had the greatest response with the PVB polymer; we consequently deduced that No.2 was Blue Nun-2, shown in Figure 7b, and No.4 was the Dadung because of the least responses with these polymers. No. 3 was then the Blue Nun-1. After four days, all frequency shifts decreased because of the decay of the polymer sensing film, but we could still discriminate each red wine. At the seventh day, we obtained the result that the sensor lost the ability to discriminate because of the decay of the polymer sensing film, as shown in Figure 7c.

In summary, on comparison of these three times, on the first day, the results showed that the discriminating ability of the SAW sensors was more significant than on the fourth day. At the fourth day, the frequency shift was less than on the first day because the ability of the polymer sensing film to adsorb VOC decreased. As shown in Figure 7, we observed that the use of PS as a detecting material failed to differentiate these red wines because PS has a benzene structure that does not readily adsorb any compound. As a result, the lifetime of a polymer sensing film was about four days; after this period the sensor gradually became insensitive to red wines. We could remove the polymer sensing films and recoat with new ones to reuse the SAW chips. In the future, we shall undertake relevant research about the lifetime of polymer sensing.

## 4. Conclusions

Our TD-GC-MS system successfully identified the marker VOC that could distinguish each wine. The marker VOC of King included methyl acetate, 2-pentanone, dimethyl disulfide, 2-methylpropyl acetate and 2-pentanol and Blue Nun-1 had 2,3-butanedione as a marker.

After determining these marker VOCs, we found a suitable polymer that readily adsorbed these compounds and applied our sensor to distinguish the red wines made from black queen, cabernet sauvignon and merlot grape varieties. As a result, PVB was appropriate to differentiate merlot from the others; PVA was suitable to discriminate black queen from the others. To enhance the credibility of our experimental results, we conducted blind tests on the red wines. We used our sensors to distinguish them within four days. After four days, our sensor gradually lost the ability to discriminate because of the decay of the polymer sensing films.

## Figures and Tables

**Figure 1 micromachines-10-00725-f001:**
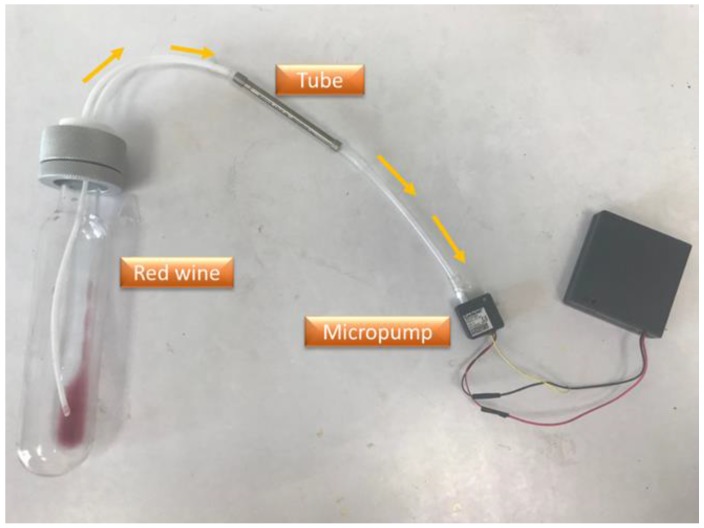
Method of extraction of a gaseous compound.

**Figure 2 micromachines-10-00725-f002:**
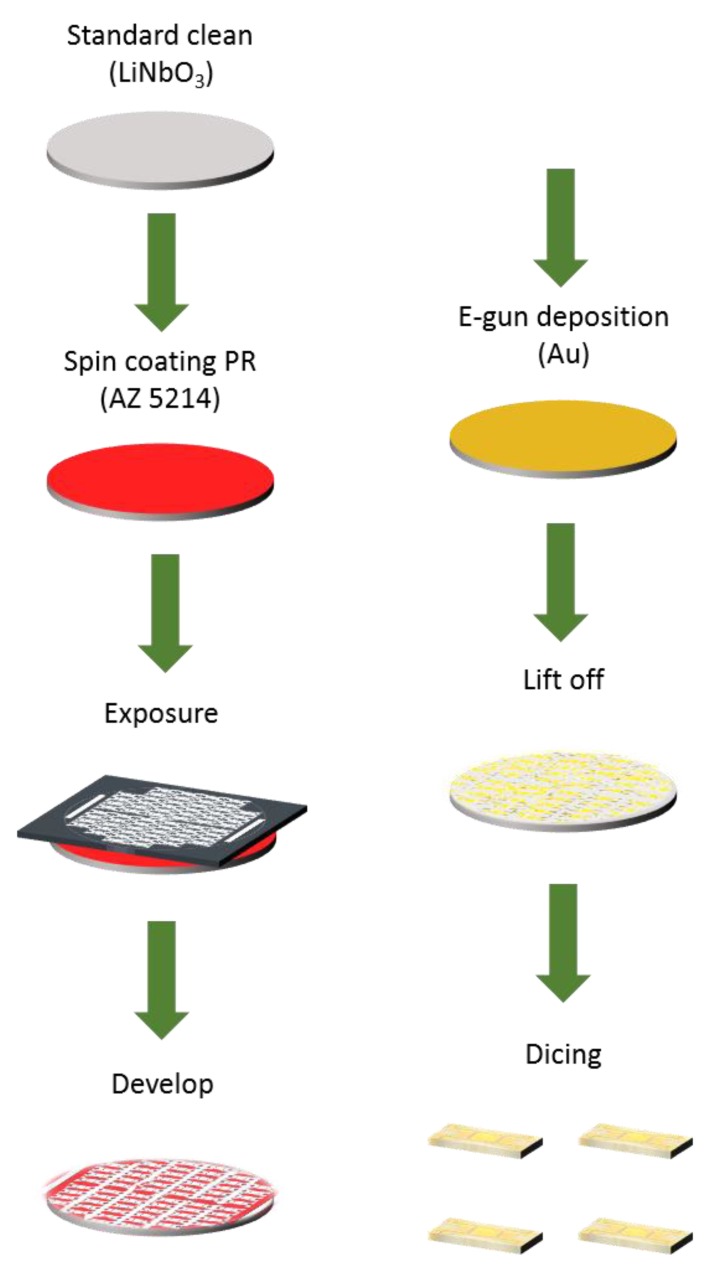
Fabrication of surface acoustic wave (SAW) chips.

**Figure 3 micromachines-10-00725-f003:**
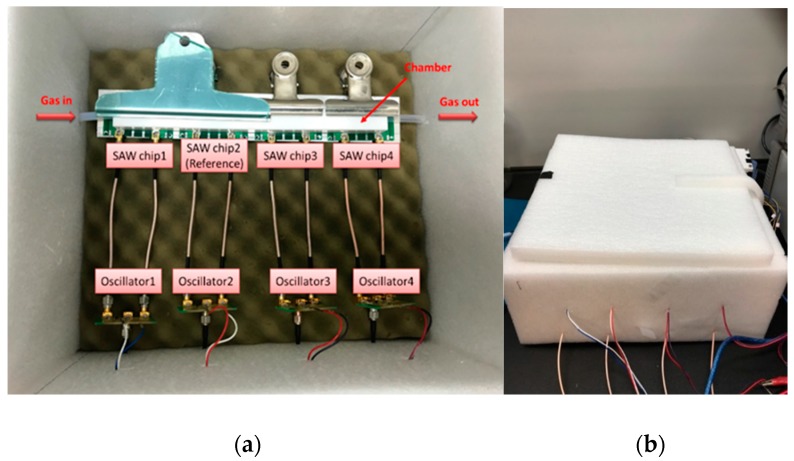
(**a**) Four-port SAW sensors and (**b**) isolation sensing chamber.

**Figure 4 micromachines-10-00725-f004:**
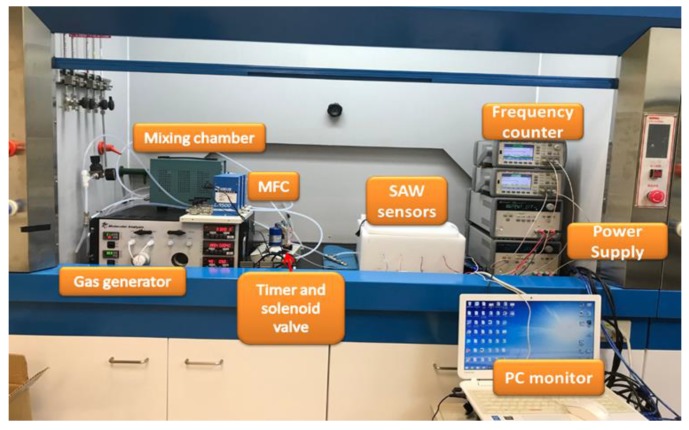
Experimental setup for gas sensing.

**Figure 5 micromachines-10-00725-f005:**
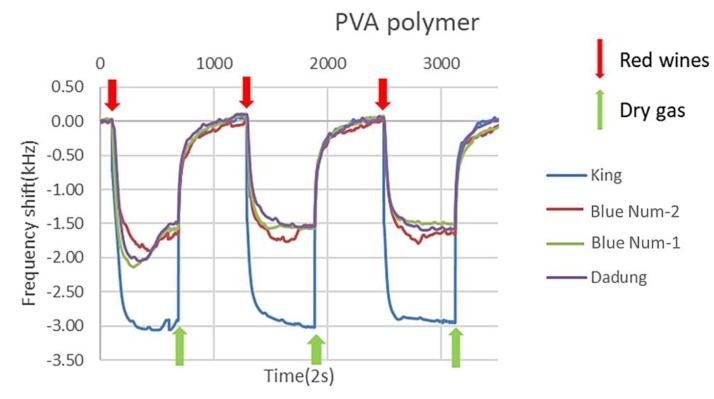
Polyvinyl alcohol (PVA) as a sensing material produces these adsorption and desorption curves.

**Figure 6 micromachines-10-00725-f006:**
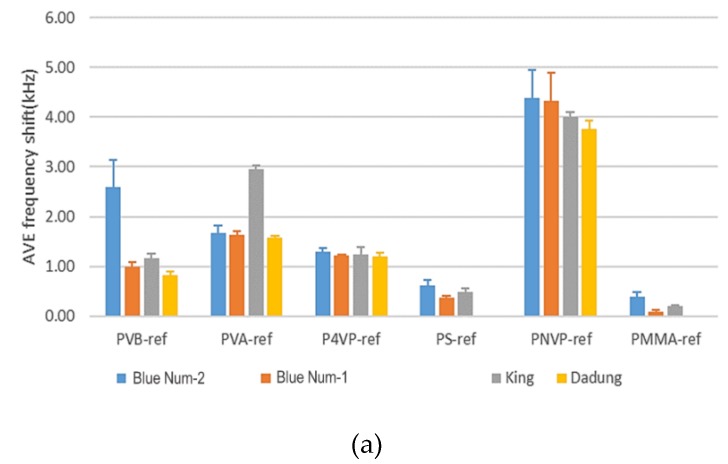
Average frequency shift of SAW sensors coated with six polymers; (**a**) first time, (**b**) second time and (**c**) third time.

**Figure 7 micromachines-10-00725-f007:**
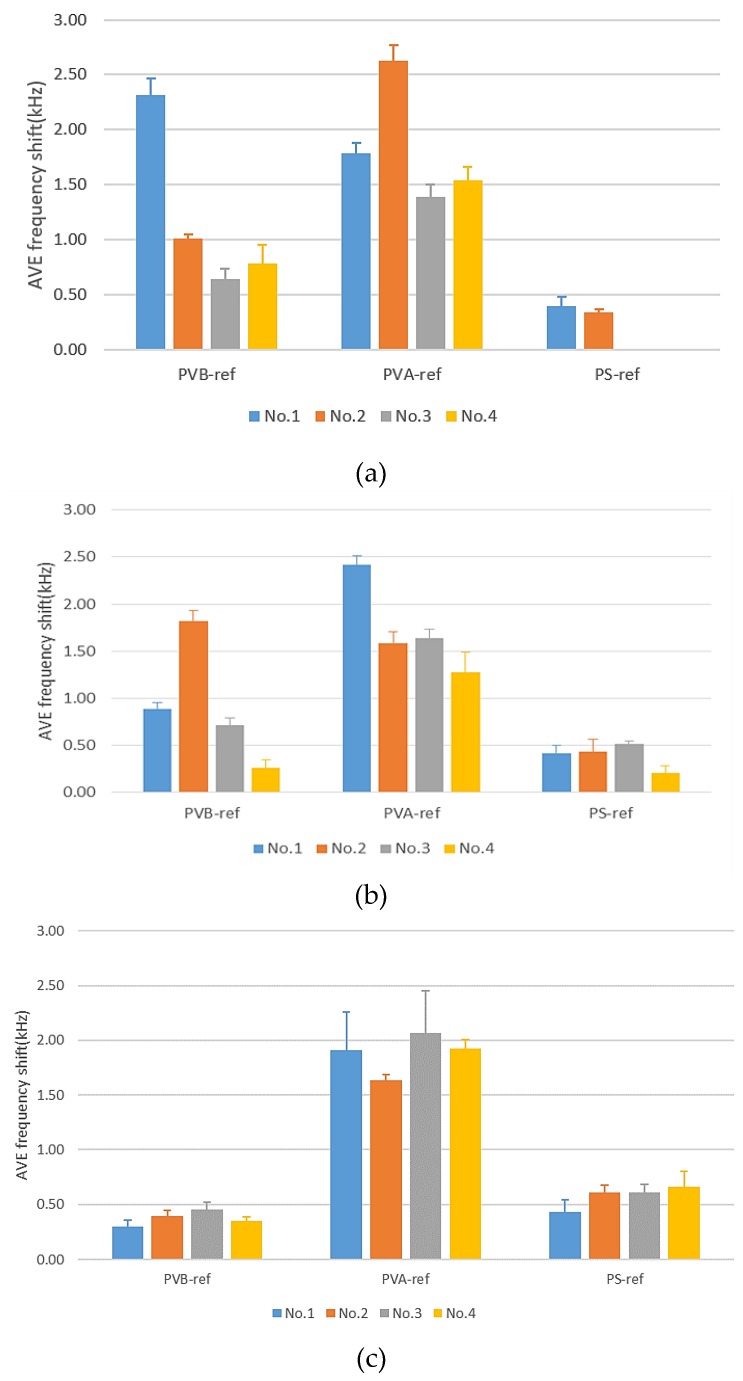
Blind test of four red wines: (**a**) first day, (**b**) fourth day and (**c**) seventh day.

**Table 1 micromachines-10-00725-t001:** Comparison of commercial gas sensors [8].

Sensor Type	Semiconductor	Infrared	Quartz Crystal	SAW
Sensor size	0.3 × 0.3 mm^2^	6.35 × 3.18 mm^2^	Diameter: 25 mm	3 × 2 mm^2^
Measured concentration	200 ppm	~ppm	0.1–100 ppm [12]	0–250 ppb
Response time	100 s	<5 s	<1 s	~100 s
Operating temperature	200–500 °C	−10–50 °C	0–40 °C	~23 °C
Mechanism	resistance	absorption of infrared radiation	mass loading	mass loading
Common detection	CH_4_, CO, C_2_H_4_, C_2_H_2_, NO, CO_2_, CH_2_CHCl, EtOH	CH_4_, CO_2_, hydrocarbons	organic or inorganic film layers	VOC, H_2_, H_2_O, H_2_S, CO, CO_2_
Disadvantages	operation at high temperature	easily influenced by temperature and humidity	interface electronics and detection range less than for SAW	interface electronics

**Table 2 micromachines-10-00725-t002:** Information of red wine samples.

	Wine Brand	King	Blue Nun-1	Blue Nun-2	Dadung
Grape Variety					
Black queen		●			●
Cabernet sauvignon			●		●
Merlot				●	

**Table 3 micromachines-10-00725-t003:** Measurement by GC–MS from DB-624 results. (Unit: 10^5^).

Markers	King	Blue Nun-1	Blue Nun-2	Dadung
Sulfur dioxide	–	1.70	2.59	–
Acetaldehyde	7.10	–	–	2.53
Acetic acid, methyl ester (methyl acetate)	85.22	–	–	–
Acetic acid, 2-methylpropyl ester (2-methylpropyl acetate)	50.12	–	–	–
Ethanol	20711.84	16530.75	22560.19	13886.27
Ethyl acetate	4872.73	1777.45	2092.90	2682.81
1-Propanol	493.72	173.19	161.85	31.20
1-Propanol, 2-methyl-	1861.24	686.80	994.77	230.52
1-Butanol	35.59	12.89	9.03	–
2-Pentanone	86.84	–	–	–
Propanoic acid, ethyl ester	82.78	25.08	63.20	11.02
Propanoic acid, 2-methyl-, ethyl ester	107.96	17.38	24.05	9.47
Disulfide, dimethyl	10.85	–	–	–
1-Butanol, 2-methyl-	70.69	90.09	77.93	53.63
1-Butanol, 2-methyl-, acetate	18.40	31.53	29.84	–
1-Butanol, 3-methyl-	2426.55	3475.05	4362.63	562.89
1-Butanol, 3-methyl-, acetate	1271.36	1408.16	1726.33	316.59
Butanoic acid, ethyl ester	36.87	3.82	4.17	84.44
Butanoic acid, 2-methyl-, ethyl ester	49.27	7.24	8.31	–
Butanoic acid, 3-methyl-, ethyl ester	54.07	268.77	225.24	–
Hexanoic acid, ethyl ester	–	21.39	31.99	–
Quinoline, 6-ethoxy-1,2-dihydro-2,2,4-trimethyl-	–	–	48.16	–

**Table 4 micromachines-10-00725-t004:** Measurement by GC–MS from DB-WAX results (Unit: 10^5^).

Markers	King	Blue Nun-1	Blue Nun-2	Dadung
Ethyl acetate	2978.03 ± 0.01	146.87 ± 236.22	1234.08 ± 373.10	456.69 ± 40.01
Ethanol	35789.09 ± 7854.27	3646.95 ± 8053.27	38669.95 ± 13566.44	15525.80 ± 1184.38
1-Propanol	301.46 ± 97.39	20.11 ± 104.03	132.26 ±64.69	74.65 ± 14.28
1-Propanol, 2-methyl-	1050.15 ± 273.24	85.63 ± 314.87	423.56 ± 229.19	224.30 ± 32.79
2-Pentanol	9.40 ± 5.19	–	–	–
2-Pentanone	300.27 ± 414.58	–	–	–
Propanoic acid, ethyl ester	185.54 ± 41.9	22.39 ± 87.84	39.23 ± 12.54	–
Propanoic acid, 2-methyl-, ethyl ester	55.94 ± 16.24	8.80 ± 8.14	72.49 ± 17.87	–
1-Butanol	9.46 ± 6.14	1.46 ± 7.46	8.41 ± 2.52	–
1-Butanol, 3-methyl-	1841.12 ± 1269.01	553.18 ± 4012.7	2300.56 ± 876.22	3596.22 ± 512.76
1-Butanol, 3-methyl-, acetate	–	0.65 ± 2.09	5.77 ± 4.45	–
2,3-Butanedione	–	8.10 ± 33.74	–	–

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
