# Peer review of "Discrimination of Red Wines with a Gas-Sensor Array Based on a Surface-Acoustic-Wave Technique"

_micromachines, 2019, doi:10.3390/mi10110725_

Round 1
Reviewer 1 Report
The authors utilized surface acoustic wave (SAW) sensors' array to distinguish various types of red wines. While the study is interesting, the manuscript quality is not up-to the mark. Sometimes, it is even hard to follow the text. For example, in the first paragraph of the introduction, its not clear what the authors are trying to say. I strongly suggest to re-write the manuscript that requires significant improvement in grammar and presentation structure. Moreover, the references are not properly included. For example, no reference is included in Table 1. In some places, it is also necessary to improve the scientific quality of the manuscript. For example, in at the start of Section 2.4 it's written that ''The fundamental principle of a SAW sensor is that a piezoelectric material becomes deformed 108 through the piezoelectric effect; a surface-acoustic wave is stimulated with a frequency from MHz to 109 GHz. When a compound passes through the sensing area, which is a coated polymer such as PVB or 110 P4VP, these compounds become adsorbed and cause a frequency shift.'' This is really vague. You can see the following paper for better explanation of SAW operation.
https://pubs.rsc.org/en/content/articlehtml/2016/ra/c6ra03148j
Similarly, as a reader I also found it hard to extract information from experimental and result sections.
Overall, I think the manuscript should completely be rewritten with better English and presentation structure. I would be happy to review it again after that.
Author Response
The authors utilized surface acoustic wave (SAW) sensors' array to distinguish various types of red wines. While the study is interesting, the manuscript quality is not up-to the mark. Sometimes, it is even hard to follow the text. For example, in the first paragraph of the introduction, its not clear what the authors are trying to say. I strongly suggest to re-write the manuscript that requires significant improvement in grammar and presentation structure. Moreover, the references are not properly included. For example, no reference is included in Table 1. In some places, it is also necessary to improve the scientific quality of the manuscript. For example, in at the start of Section 2.4 it's written that ''The fundamental principle of a SAW sensor is that a piezoelectric material becomes deformed 108 through the piezoelectric effect; a surface-acoustic wave is stimulated with a frequency from MHz to 109 GHz. When a compound passes through the sensing area, which is a coated polymer such as PVB or 110 P4VP, these compounds become adsorbed and cause a frequency shift.'' This is really vague. You can see the following paper for better explanation of SAW operation.
https://pubs.rsc.org/en/content/articlehtml/2016/ra/c6ra03148j .
Similarly, as a reader I also found it hard to extract information from experimental and result sections.
Overall, I think the manuscript should completely be rewritten with better English and presentation structure. I would be happy to review it again after that.
Revision:
Thanks for your review comments. I have improved the entire manuscript by professional native editor.
We have added citation to table 1. We have added citations to further explain the pros and cons of other commercial sensors from line 62-68. In page 4, line 122-127, we have referred to the paper you gave and rewrite the explanation of working mechanism of SAW to be more clearly. Thank you for your suggestion. (https://pubs.rsc.org/en/content/articlehtml/2016/ra/c6ra03148j )

Reviewer 2 Report
The work contains elements of scientific novelty and features of the application. However, it should improve significantly at this stage. Below are my remarks and comments:
page 1, line 35-38, Please comment on what the sentence refers to:
"An electronic nose and a human panel were used for the quantification of wines formed by binary mixtures of various red grape varieties at varied percentages. Among these factors, the grape variety mainly causes the individual flavor of each red wine".
Whether for authors 'research or other researchers' research, but then you need citations.
page 2, line 54-66 I believe that the authors treated the use of various types of sensors or sensor matrices to analyze compounds from the VOC's group quite briefly. Each sensor has its advantages and disadvantages and please include this in a new paragraph on this subject. I also suggest that you read the literature and quote it:
1) Review of Portable and Low-Cost Sensors for the Ambient Air Monitoring of Benzene and Other Volatile Organic Compounds, Sensors 2017, 17(7), 1520
2) Currently Commercially Available Chemical Sensors Employed for Detection of Volatile Organic Compounds in Outdoor and Indoor Air, Environments 2017, 4(1), 21
3) Application of electrochemical sensors and sensor matrixes for measurement of odorous chemical compounds. Trac Trends Anal. Chem. 2016, 77, 1–13.
4) Applications and advances in electronic-nose technologies. Sensors 2009, 9, 5099–5148.
On “Electronic Nose” methodology. Sens. Actuator B Chem. 2014, 204, 2–17
page 2, line 63-66, which is a premise that only a SAW sensor is suitable for wine analysis ?, This text does not show, please expand this paragraph.
page 4, equation 1, please enter the units for each symbol in this equation.
page 6, line 181-182, Why was helium used as gas? after all, in real conditions it would be best to use air. There will be technical difficulties with the use of helium in measurements with the use of the SAW sensor.
page 11, line 290-292, What will the lifetime of the SAW sensor be then? since after four days his tendency drops rapidly!
Conclusion - please refer to the results of other work, are the markers obtained characteristic for red wines ??
Round 2
Reviewer 1 Report
The revised version reads much better than the original one. However, I still think the English should be further improved before it can be published. Thanks.
Reviewer 2 Report
I believe that the amendments introduced are satisfactory and I recommend it for further stages of evaluation.